# Using remarkability to define coastal flooding thresholds

Frances C. Moore[1]* & Nick Obradovich[2]

Coastal flooding is increasingly common in many areas. However, the degree of inundation and associated disruption depend on local topography as well as the distribution of people, infrastructure and economic activity along the coast. Local measures of flooding that are comparable over large areas are difficult to obtain. Here we use the remarkability of flood events, measured by flood-related posts on social media, to estimate county-specific flood thresholds for shoreline counties along the east coast of the United States. While thresholds in most counties are statistically-indistinguishable from minor flood thresholds of nearby tide gauges, we find evidence that several areas experience noticeable flooding at tide heights lower than existing flood thresholds. These 22 counties include several major cities such as Miami, New York, and Boston, with a total population over 13 million. Our analysis implies that large populations might currently be exposed to nuisance flooding not identified via standard measures.

[1] Department of Environmental Science and Policy, University of California Davis, Davis, CA, USA. [2] Max Plank Institute for Human Development, Berlin, Germany. *email: fmoore@ucdavis.edu

Coastal floods and inundation are projected to produce some of the primary social impacts of climate change, imposing significant costs on communities around the world[1–3]. Flooding due to high tides, storm surges, or a combination of the two is increasingly common in many coastal areas and is projected to become more frequent and severe as sea-levels rise globally[4–6]. Understanding where coastal floods happen, identifying which meteorological and tide conditions produce floods, and grasping the consequences for flood-affected communities and infrastructure is critical for coastal flood preparation and response[3].

Understanding the drivers and consequences of coastal flooding can be challenging, however, because the extent of flooding may be highly variable within a small geographic area, depending on local topography and bathymetry[7]. Moreover, for planning and response purposes, the effects of flooding on residents and businesses are arguably more important than the simple geographic extent of flooding[8,9]. The same degree of inundation could have substantively different social impacts, depending on the distribution of people, infrastructure and economic activity along the coast. Measuring the effects of floods therefore requires highly localized measures of inundation and the effects of that inundation for coastal communities[8,10].

Despite the importance of localized information on the extent of coastal flooding and its impacts, gauge stations measuring tide heights are sparse. For example, along the 3700 miles of coast making up the eastern and Gulf seaboards of the United States, there are only about 132 tidal gauge stations with long-term records. Further, translating tide heights into local inundation is not straightforward. Most gauges have three tide heights associated with minor, moderate or major flooding. However, the frequency of exceeding these thresholds varies widely—the fraction of days experiencing minor flooding since 2004 ranges from 25% in Wilmington, NC, to 0% in Bar Harbor, ME. Moreover, the severity of the different types of flooding is also not well standardized across gauges. A moderate flood at one gauge might imply very different consequences from a moderate flood at a different gauge[11].

Here, we use geolocated social media data with high temporal resolution to estimate local coastal flood thresholds[12]. We propose using the remarkability of a particular high-tide event, as measured by the volume of tweets about flooding generated in a particular day, as a measure of flood occurrence and severity. Other scholars have used social media data to identify damage[13,14] and aid management[15,16] of severe natural disasters, such as earthquakes[17–19], heat waves[20], hurricanes[21,22], snowstorms[23], and wildfires[24]. Researchers have also recently examined the ability to use social media to detect public attention paid to other climatic factors[25]. Although there have been case studies using text and pictures on social media to map inundation for specific inland flood events[26,27], this paper presents a novel, general method for assessing the severity of regular coastal flood events across a wide geographic area.

There are two principle benefits of our approach to measuring floods. Firstly, because of the wide geographic coverage and relatively high density of Twitter data, we are able to estimate localized (i.e., county-specific) flooding thresholds, rather than relying on extrapolation from a sparse network of tide gauges. Secondly, our remarkability metric naturally integrates a measure of the social consequences of flooding, which is theoretically standardized within a particular county and time period. A flood that covers an important roadway will be more remarkable than one of the same extents that only covers farmland. Similarly, a flood in a highly populated part of a county will affect more people and be more remarkable than a similar flood in sparsely populated area. Thus, these social media derived flooding thresholds implicitly integrate information on the distribution of people and infrastructure along the coast and the vulnerability to flooding at different tide heights. As such they can complement existing flood planning tools. In particular, social media might provide a sensitive instrument to measure nuisance coastal flooding that is both more regular and less consequential than the flooding covered by other tools, such as the FEMA flood maps of 1 in 100 and 1 in 500 year floodplains.

## Results

**Changes in flood frequency.** Figure 1a shows flood frequency since 2004 for each tide gauge in the dataset. Neighboring gauge stations tend to experience flooding at similar times, likely due to nearby areas being affected by the same meteorological events. Figure 1b shows the average monthly number of floods, with evidence for a steady increase in the average number of flood events over time. Noticeably though, flood frequency differs substantially across the different gauges (Fig. 1a). During the primary period of analysis (March 2014 to November 2016), the gauge in Wilmington, NC, recorded over 150 flood days, or over 4 per month on average, whereas other stations experienced only one or two over the whole period. This variation likely reflects real differences in the susceptibility to coastal flooding across regions, but may also reflect idiosyncratic variation in the determination of flooding thresholds. It is possible that tide heights above the minor flood threshold are more consequential in places with that experience them rarely than in places that experience them frequently, even though both events might be formally classed as minor floods.

**Flood tweets and tide height.** We first combine data on social media posts about flooding with data on weather and tide heights. The starting set of social media data is all Twitter posts geolocated within shoreline counties along the Atlantic and Gulf coasts of the United States between March 2014 and November 2016. Tweets about flooding are classified using a simple bag-of-words approach, where any post containing at least one word or phrase identified as possibly referring to flood events was labeled as a flood tweet (see "Methods" section). We aggregate the total number of tweets, the total number of flood tweets, and the number of Twitter users to the county-by-day level.

We combine these social media data with tide gauge data on maximum daily tide height from active tidal gauge stations along the Atlantic and Gulf coasts active since at least 2009. Local flood thresholds for minor, moderate, and major floods were obtained from NOAA's Advanced Hazard Prediction System (AHPS)[28] and, where no match could be found, were estimated based on approximations used in ref. [11]. We matched each county to the closest tide gauge based on the population-weighted centroid of the county. We also add control variables for total daily rainfall (4th order polynomial) and cumulative 5-day rainfall (quadratic)[29]. The resulting dataset has 413,000 observations from 237 counties, and includes 473,000 flood-related tweets from a population of over five million twitter users.

Figure 2 shows evidence from the full dataset, aggregated to the weekly level, that across all counties in the sample, flood-related tweets do respond to objective metrics of the magnitude of inundation (measured by maximum tide height and local flood threshold). The estimated average response across the sample shows a small threshold effect at the minor flood threshold, with a steep increase in the number of tweets above that threshold. This specification includes county, state-by-month and year fixed effects (dummy variables) and controls

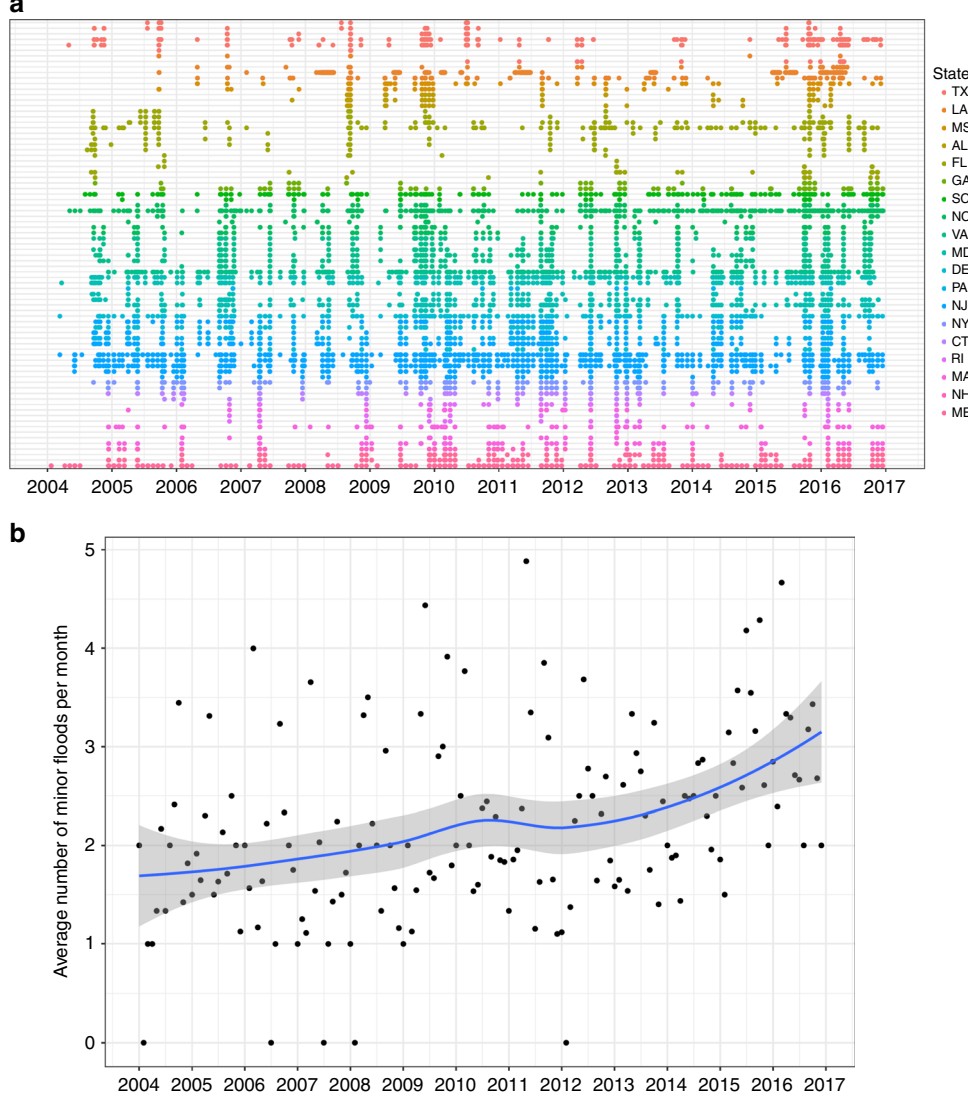

**Fig. 1 Geographic and temporal patterns of tidal flooding in the Atlantic and Gulf Coasts of the United States. a** Representation of flood frequency for 83 tidal gauge stations along the Atlantic and Gulf coasts of the United States. Each row is a different gauge and dots show days when maximum tide height exceeded minor flood stage for that gauge. Gauges are ordered geographically from north to south along the Atlantic coast, and then east to west along the Gulf coast. Colors show the state of each gauge. **b** Average number of days per month with maximum tide height exceding minor flooding threshold for the 83 gauges shown in **a**. The line shows a loess regression through the data with 95% confidence interval.

for cumulative 5-day precipitation (quadratic) and the number of Twitter users in a county-week. The functional form and additional details are given in the "Methods" section. Our subsequent analysis examines data from each county independently and identifies a county-specific threshold that best explains the pattern of flood-related posts.

**County-specific flood thresholds**. In order to determine county-specific flooding thresholds using kinks in the relationship between flood-related tweets and maximum tide height, we perform a model selection exercise for each county using daily data (see "Methods" section). We use the number of daily flood-related tweets as a county-specific measure of the seriousness of coastal flooding in a particular county and day. For each county we fit a local response function relating the number of flood-related tweets and maximum daily tide height of the following form:

$$y_{\mathbf{cd}} = \beta_c + \beta_1 I(h_{\mathbf{cd}} > A_c) + \beta_2(h_{\mathbf{cd}} - A_c) \times I(h_{\mathbf{cd}} > A_c) + X_{\mathbf{cd}},$$

$$(1)$$

where $y_{\mathrm{cd}}$ is the number of tweets in county c on day d, $h_{\mathrm{cd}}$ is the maximum daily tide height and $X_{\mathrm{cd}}$ is a vector of controls including polynomial functions of 5-day cumulative precipitation (2nd order) and daily rainfall (4th order), the total number of Twitter users in that county and day, and month fixed effects to flexibly control for county-specific seasonal effects. $\beta_c$ is a county-specific intercept and $I(h_{\mathrm{cd}} > A_c)$ is an indicator variable taking a value of one if daily tide height exceeds a threshold value of $A_c$ and 0 otherwise. Therefore $A_c$ is a county-specific threshold, above which flood-related tweeting increases linearly with tide height. We find changepoints ($A_c$) for each county that best fits the observed relationship between flood posts and tide height using the minimum AIC criterion, searching between the 50th and 99th percentile of observed tide height in 2-cm intervals. Given the dependent variable is a count variable, all regressions are fit as negative binomial models. To ensure sufficient data to estimate a response, we require that at least 20% of days have some posts about flooding and require that estimated coefficients are positive (consistent with the proposed interpretation of increasing

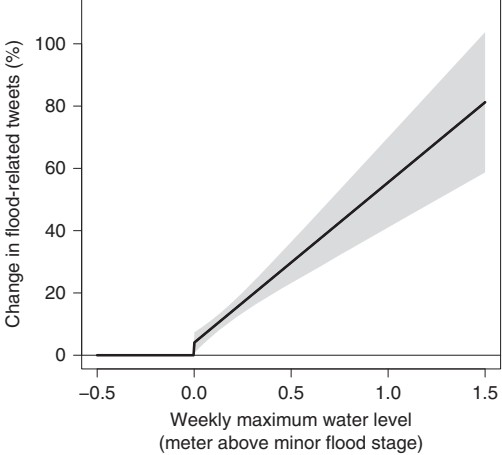

**Fig. 2 Estimated relationship between weekly maximum tide height and the % change in flood-related tweets, using the functional form given in Eq. (1).** This is estimated using data from all counties, aggregated to the weekly level. Fixed effects by county, state-month, and year are included, as well as controls for cumulative weekly rainfall (quadratic) and the number of users (see "Methods" section). Shaded areas show the 95% confidence intervals, with standard errors clustered at the state level.

remarkability), leaving 140 counties for which models were successfully estimated. Additional information on data and methods is given in the "Methods" section.

We define a remarkable threshold for coastal flooding, corresponding to a 25% increase in the volume of posts about flooding, given by the selected model for each county. Figure 3a, b gives two measures of how this threshold compares with flood thresholds of the nearest tide gauge. Figure 3a gives the absolute threshold above the mean lower low water (MLLW) datum. In general the remarkability threshold qualitatively tracks the minor flooding threshold, although on average it tends to be lower. Figure 3a shows a strong correlation between the remarkability threshold and the minor flooding threshold, but this is mostly driven by the large variation in tidal range (the difference between high and low water), which varies from <0.5 m along much of the Gulf Coast to over 3 m in Maine. Much of this variation is captured mechanically through our threshold search process. Another way of comparing our estimated remarkable thresholds with the minor flooding thresholds of nearby gauges is the probability of exceedance, shown in Fig. 3b. This accounts for differences in both tidal range and seasonal variability across locations.

**Outlier counties.** Outlier counties are identified as those where there is 95% confidence that the noticeable flooding threshold is different from that of the minor flood threshold of the nearest tide gauge (highlighted in Fig. 3). For most counties, the remarkability threshold is not statistically different from the minor flood threshold. We identify 22 counties where we estimate noticeable flooding is more common than given by the tide gauge threshold (lower right, Fig. 3b) and two counties where it is less common (upper left, Fig. 3b). In general these outlier counties have a higher fraction of their population living at low elevations compared with other coastal counties (Supplementary Fig. 1). Other characteristics of these counties are given in Supplementary Table 1.

Figure 3c maps the difference between estimated and gauge thresholds in terms of the quantiles of tide height distribution, together with the location of tide gauges used in the analysis, and the locations of identified outliers. Although outlier

counties are distributed across a number of states along the Atlantic and Gulf coasts, there are several geographic clusters that stand out. There are notable clusters around the Boston, New York, and southern New Jersey areas. These clusters are notable in that neighboring counties that may be assigned to different tide gauges or may even be in different states are identified as experiencing more frequent flooding, despite regression analysis being conducted independently by county (Supplementary Table 1). In addition, almost a quarter of outlier counties are along the Texas Gulf Coast: five of the seven Texas counties for which models were successfully estimated are identified as outliers where flooding occurs significantly more frequently than would be inferred from tide gauge measures. Finally, Florida also has a number of outlier counties distributed throughout the state, including in populous Miami-Dade and Jacksonville counties. Figure 4 shows the fitted response functions with both the noticeable tide height and the minor flood threshold for a subset of the 24 outlier counties.

The set of outlier counties also suggest there may be two reasons why flood frequency estimated through social media posts might differ from that established from nearby tide gauges. In a small number of cases, (white rows in Supplementary Table 1), counties are several miles from the nearest gauge suggesting that in some cases the density of tide gauge measurement is simply not sufficient to capture the heterogeneity of local conditions along the coast. In most cases, however, the matched tide gauge is either within the county or in a neighboring county (green rows in Supplementary Table 3), suggesting that in these cases, minor tidal flooding that is remarkable to residents happens at a tide height different from that defining minor coastal flooding.

The case of Wilmington, NC, is illustrative. Sweet et al.[11] point out that this gauge has one of the lowest flood thresholds, relative to the mean higher high water (MHHW) datum, resulting in minor flooding being recorded every 4–5 days. However, this flooding only affects one minor, low-lying, and undeveloped highway[11]. Our results suggest that flooding noticeable to residents in the area happens at 0.33 m above MHHW rather than at the 0.25 m flood threshold, resulting in remarkable floods occurring approximately once every 2 weeks or half as frequently as would be implied by the gauge threshold.

For most of the outlier counties, however, we find the opposite: that remarkable flooding is more common than would be inferred from gauge thresholds. This is particularly the case for areas along the Texas coast (Fig. 3b, c). The largest difference between estimated and established flood thresholds, in terms of probability of exceedance, occurs near Beaumont, TX, where in two neighboring counties, we independently estimate significant tide heights to occur 0.5 m below the established level. Over the relevant period (March 2014 to December 2016), tide heights above the minor flood stage are not observed at this gauge. And yet for two neighboring counties we precisely estimate response functions showing an increase in flood-related posts at much lower tide heights (Fig. 4).

Inspection of the tweets from these counties demonstrates at least ten moderately or very serious flood events over the relevant time period with residents reporting consequences, such as canceled school, unsafe driving conditions, or an inability to get to or from work. These events are almost universally associated with moderate-to-heavy rainfall, but the statistical analysis suggests that high tides contribute to drainage problems in the area, exacerbating flood conditions. Maximum tide height is a significant explanatory variable even after flexibly controlling for daily precipitation and cumulative 5-day rainfall.

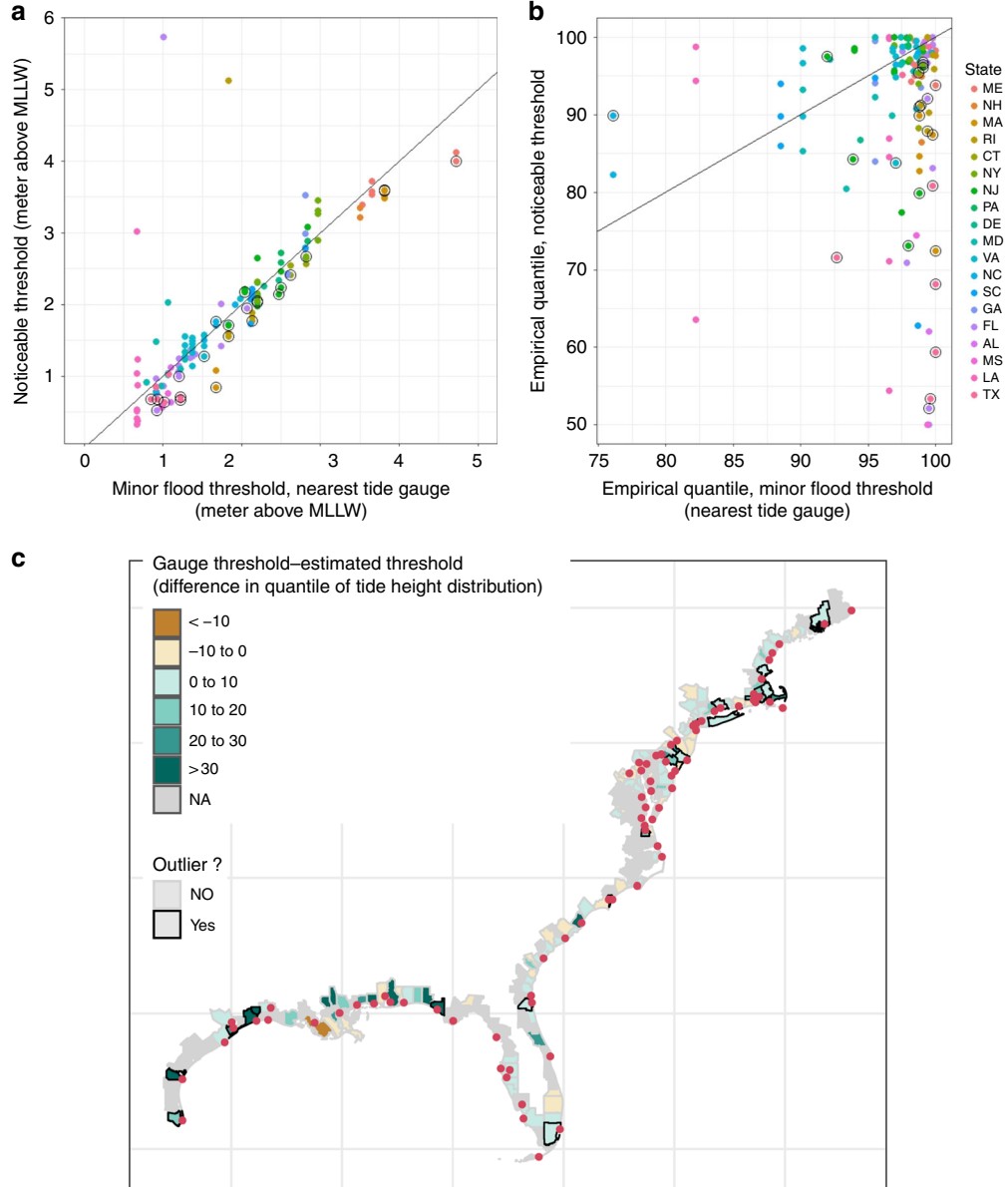

**Fig. 3 Relationship between estimated noticeable threshold (25% increase in flood-related posts) and the minor flood threshold of the nearest tide gauge.** Shown in absolute terms as meters above mean lower low water (**a** MLLW) and as quantiles of the observed distribution of daily tide heights over the period (**b**). Much of the variation in absolute tide heights (**a**) is driven by large differences in the tidal range between gauges. Circled points show 24 counties where the estimated noticeable threshold can be statistically distinguished from the minor flood threshold at the 95% confidence level. Given the number of comparisons, six false positives would be expected. **c** Map showing the study area (shoreline counties along the Atlantic and Gulf coasts of the United States) and difference between estimated noticeable flooding thresholds and the minor flood threshold of the nearest tide gauge in terms of quantiles of the tide height distribution for counties where sufficient data exist to estimate a response function. Red dots show tide gauge locations.

## Discussion

Here, we show firstly that flood-related tweets can be used to measure the remarkability of coastal flooding events. Pooling data from across the study area, we show that the volume of tweets about flooding responds in expected ways to tide height and local flood thresholds. Secondly, we identify thresholds for remarkable flooding based on best-fit, county-specific functions relating local tweets about flooding to tide height at the nearest gauge, identifying several locations where flooding appears to occur at levels different from the minor flood threshold.

Despite the fact that our models are fitted independently by county, a number of the identified counties are geographic neighbors, increasing confidence in the conclusion that flood conditions in these areas differs from that implied by gauge flood thresholds.

Inspection of the outlier counties reveals that discrepancies may arise for two reasons: counties are geographically distant from the nearest gauge or, more commonly, local flood thresholds do not reflect the average experience of flooding for residents in that county. Although there are exceptions, remarkable flooding tends to happen at lower tide heights, with implications for the vulnerability of these areas to future sea-level rise.

Some care should be taken in interpreting these findings. In particular, our remarkability thresholds are estimated using the population of Twitter users, which is a subset of the general population. Demographics of Twitter users—as compared with the general population—are not available but to the extent differences between the two groups are correlated with factors that determine whether people are affected by flooding, they will affect

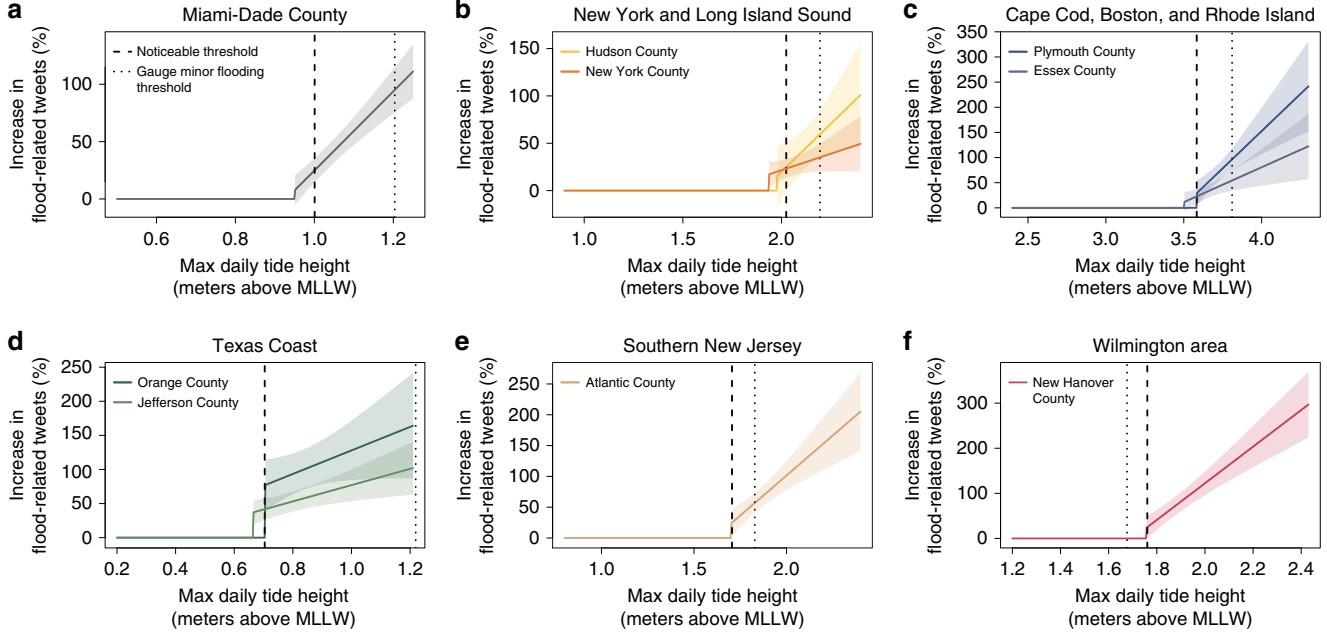

**Fig. 4 Fitted relationship between tide height and flood-related social media posts for a subset of the 24 outlier counties that represent the geographic clusters of outlier counties.** a Florida (represented by Miami-Dade county) (**b**) New York and Long Island Sound (**c**) Boston, Cape Cod and Rhode Island (**d**) the Texas Coast (**e**) southern New Jersey, and (**f**) Wilmington, NC. Vertical lines show the minor flood threshold of the nearest tide gauge (dotted line) and the estimated noticeable flooding threshold corresponding to a 25% increase in flood-related tweets. Shaded areas show the 95% confidence interval.

the generalizability of results beyond the population of Twitter users. In addition, we note that the number of Twitter users in a county is likely related to the variance of the estimated response functions—counties with more people Tweeting produce more data and therefore may have a more precisely estimated response. We do find evidence for this effect (Supplementary Table 2), but no evidence that the number of Twitter users in a county systematically biases estimated thresholds in either direction. This does mean that rural areas are less likely to be identified as outliers in our analysis due to higher variance estimates resulting from fewer Twitter posts in rural areas.

In addition, we note that one potentially confounding effect is the changing remarkability of flood events with repeated exposure. Moore et al.[25] showed that temperature anomalies become less remarkable when experienced repeatedly over a period of 2–8 years. It may be the case that a similar effect applies to nuisance flooding, such that the same level of inundation gradually evokes a smaller response on Twitter as it becomes less and less surprising. Supplementary Fig. 2 shows some suggestive evidence across the whole sample that this is the case: minor floods (as defined by tide gauge thresholds) produce a larger response when the last flood was a long time ago compared with when it was more recent. This result should be interpreted with caution though as it is not clear whether the effect is driven by normalization (floods in places where they happen regularly are simply less remarkable) or adaptation (places with regular floods have adjusted to minimize the impact of flooding).

Our analysis has also demonstrated the potential role that remarkability metrics derived from social media data could play in understanding the significance of specific hydrometeorological events. Because they are temporally continuous and spatially dense, social media data potentially allow for a more localized mapping of impacts than may be possible with standard data sources. In addition, these data integrate over multiple pathways by which residents are affected, which may differ from location to location, and measure a more standardized impact, in terms of social consequences, than metrics based exclusively on physical

thresholds. For both these reasons, impact thresholds based on remarkability may be particularly useful for comparing the sensitivity of communities across large geographic areas, in addition to being of use to local planners and communities.

## Methods

**Data sources and processing**. Twitter data are all tweets geolocated within 235 shoreline counties along the Atlantic and Gulf coasts of the United States between March 2014 and November 2016. The sample contains Tweets from over five million unique users. The number of Twitter users steadily increases over time, with a sharp drop in late 2014 likely associated with a change in Twitter's policy on geolocating Tweets (Supplementary Fig. 3). The number of Twitter users is included as a control variable in all regressions.

Tweets about flooding were identified using a simple bag-of-words approach where any tweet containing one of the following phrases was identified as being about coastal flooding:

flood, floods, flooded, flooding, flood hazard, aquatic hazard, storm drain, storm drains, stormdrain, stormdrains, subside, subsiding, drain, drains, drainage, rising waters, rising water, crest, crested, waters have risen, water has risen, waters rose, water level, river rose, sea rose, tide, tidal, record high, sandbag, sandbags, high water, high waters, covered by water, water level, doused, douse, drenched, drench, drenches, inundate, inundated, inundates, low-lying, low lying, low elevation.

The error rate in this classification system and the implications of those errors for our conclusions were systematically assessed through a manual validation of all 3305 tweets identified as being flood related from three counties randomly selected from the set of outlier counties. This analysis indicates that while the false-positive rate is high (64%), this error is not correlated with tide height and therefore is unlikely to bias estimates in our analysis (additional discussion Supplementary Table 3). The total number of tweets, total number of flood-related tweets, and total number of Twitter users are aggregated to the county-by-day level.

Tide data for 132 gauge stations along the Atlantic and Gulf coasts were obtained from NOAA. Gauges had to be active between 2014 and 2016 and have a record dating back to 2009. Following Sweet and Park[28] flooding thresholds for each gauge were obtained from the NOAA's AHPS. For gauges where the closest AHPS station was more than one mile away, approximations based on Sweet et al.[11], in which flooding thresholds are defined based on the great tidal range at the gauge (difference between MHHW and MLLW), were used.

Each county was mapped to the tide gauge that was nearest to the population-weighted centroid of the county[30], resulting in 82 stations used in the analysis. Daily total and cumulative 5-day rainfall were calculated for each county as a spatial average of PRISM data[29].

**Definition and estimation of noticeable flood thresholds**. To estimate an aggregate, average response function across the whole sample (Fig. 2), data from all

counties are pooled and aggregated to the weekly level. At the weekly level, there is a sufficient number of flood-related tweets that the variable can be treated as continuous and the regression estimated using OLS (Eq. 2).

$$\mathrm{asinh}(y_{cw}) = \beta_c + \beta_1 I(h_{cw} > F_c) + \beta_2 (h_{cw} - F_c) \times I(h_{cw} > F_c) + X_{cw} \quad (2)$$

With variables defined as described in the main text except all variables are at the weekly (w) instead of daily (d) level and the threshold for each county is the minor flood stage of the nearest tide gauge ($F_c$). The inverse hyperbolic sine transformation is similar to a log transformation for nonnegative variables except it allows zeroes to be retained in the analysis[31]. The vector of control variables ($X_{cd}$) includes a quadratic in cumulative 5-day precipitation, the inverse hyperbolic sine of the number of Twitter users in that county and week, and fixed effects (dummy variables) for county, state-month, and year. Collectively these fixed effects flexibly control for all time-invariant differences between county, state-specific seasonal effects, and common time trends. Standard errors are clustered at the state level, accounting for within-state spatial and temporal autocorrelation.

To estimate county-specific flood thresholds, analysis is done at the daily level. For each county with a sufficient density of flood-related tweets (at least 25% of days with more than 0), a negative binomial model is fit using the formula given in Eq. (1). Negative binomial models account for the discrete nature of the number of flood-related posts at the daily level, while allowing for overdispersion of the data. All thresholds between the 50th and 99th percentile of observed tide heights are tested, in 2 cm increments. The model with minimum AIC is selected, after constraining the set of models to those with positive coefficients for the tide height coefficients. This leaves 140 counties for which there are sufficient data where models are successfully identified.

The noticeable flood threshold for each county ($T_{c\_25}$) is defined as the threshold for the selected model ($A_c^*$) if the discontinuity at the threshold is >25%, or the point above the threshold where flood-related tweets increase by 25% (Eq. 3)

$$T_{c\_25} = \max\left(A_c^*, A_c^* + \frac{0.25 - \beta_1}{\beta_2}\right) \quad (3)$$

Standard errors for $T_{c\_25}$, used to identify outlier counties where $T_{c\_25}$ is statistically different from the minor flood threshold, are calculated using the delta method[32].

**Reporting summary**. Further information on research design is available in the Nature Research Reporting Summary linked to this article.

## Data availability
Source data to generate all manuscript figures are provided with the manuscript. Tide gauge data are from https://tidesandcurrents.noaa.gov/. Precipitation data are from PRISM http://www.prism.oregonstate.edu/. We collected our Twitter data from the public domain in adherence with Twitter's Developer Agreement. Twitter data are restricted from public redistribution by the Twitter terms of service. Raw Twitter data may be procured through Twitter's GNIP service.

## Code availability
Code and data to reproduce all figures in the manuscript is available at https://franmoore. faculty.ucdavis.edu/publications.

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

## Author contribution
F.C.M designed the study, performed the data analysis, and wrote the paper. N.O. provided data and wrote the paper.

## Competing interests
The authors declare no competing interests.
