## [Peer Review File · Nature Communications]

Reviewers' comments:

Reviewer #1 (Remarks to the Author):

General Comments

The contribution is interesting and well-written, although lacking in detail sufficient for publication. Additionally, there are issues with the methodology that would require clarification and perhaps recomputation of results (described below). Fundamentally, as the results are largely congruent with those of existing methods based on minor flood frequency, the work does not reach the level of *extreme importance to scientists in the specific field*. Therefore, my recommendation is that this work is not suitable for publication in Nature Comm., but may be well-received by a specialist journal after addressing some core issues.

Specific Comments

1) I do not recall seeing any normalisation or consideration of the varying number of Twitter users over the period of record.

2) Mention is made of "grasping consequences", "coastal flood preparation and response", yet nowhere is FEMA and its flood maps mentioned. As those are the ultimate insurance-based reference for "response" to the community, it seems that these maps should be considered.

3) Line 15:

Flood thresholds based on remarkability can provide useful information to complement existing measures of flood frequency.

-- Thus assertion seems reasonable, but is speculative. A clear quantitative demonstration is needed to support such a statement.

4) Line 65-67, 236-243

Tweets about flooding are classified using a simple "bag-of-words" approach, where any post containing at least one word or phrase identified as possibly referring to flood events was labeled as a flood tweet (see Supplementary Information).

-- I did not find a Supplement, but this seems to be in Methods.

-- The list of words to identify flood-relevant tweets is ad-hoc, and I do not recall any sensitivity study to identify the relative importance and impact of specific, or subsets of words on the results.

-- These specific words: [army corps of engineers, levee, levees, levy, deep water, shallow water, saturate, saturated, saturates, soak, soaked, soaks, puddle, coastal, splash, splashed, splashes, spray, sprays, sprayed, got wet, get wet, rising sea, sea-level, sea-level-rise, sea level, SLR] are too general to be considered flood-specific.

5) Figure 1 is unconvincing as a demonstration of "a steady increase in the average number of flood events over time".

Reviewer #2 (Remarks to the Author):

This work discusses an approach to the estimation of coastal flooding threshold that is based on an analysis of social media data. In particular, authors learn new local flood threshold based on the increase in number of flood-related tweets. They then compare their thresholds with those typically used and derived from flood gauges. Results show that some differences exist between social media derived and traditional flood thresholds. For some counties, authors claim that the thresholds derived from social communications might be better representatives of the actual impact of flood events. An analysis of the aforementioned differences also highlights some areas

where the density of flood gauges should be increased, thus possibly contributing to drive future interventions.

The topic of this work is interesting and timely and the paper is well-written and easy to follow. In addition, the figures are clear, nice, and very informative. From the technical viewpoint however, the work is a bit shallow. Authors could have carried out additional experiments, as detailed in the following, to better evaluate their system/thresholds and avoid possible biases.

Please, find below a detailed list of concerns and suggestions:

- The initial claims of the Introduction section (e.g., flooding is increasingly common) would be more convincing with supporting references, in addition to Supplementary Figure 1.
- An important pioneering reference on the use of social media data for earthquake management, that authors might consider citing, is the following:
Earle, Paul. "Earthquake twitter." *Nature Geoscience* 3.4 (2010): 221.
- Among the works leveraging social media data for assessing the extent of damage, it might be worth citing:
Avvenuti, Marco, et al. "Nowcasting of Earthquake Consequences Using Big Social Data." *IEEE Internet Computing* 6 (2017): 37-45.
- In explaining results for the noticeable thresholds, it would be very useful to have a geo map of the discrepancy between noticeable thresholds and gauge thresholds (with also markers for gauge positions). This would help to better assess the conclusions drawn by the authors.
- I am also interested in assessing whether the discrepancy might be due to some bias. For instance, is it possible that counties with lower (Twitter) population could result in bigger discrepancies? Authors should account for this factor (and possibly other factors) when presenting their results.
- From the technical viewpoint, this work is rather simple. Other recent works carried out more detailed analyses on social media-derived data, for example by also considering the content of shared messages to identify expressions of fear or mentions of damage. Interestingly, these works demonstrated the usefulness of such in-depth analyses. Authors of this work however, only take into account the volume of flood-related tweets.
- In my view, the system proposed in this work is highly dependant on the "sensitivity" of the population in a given area to flood events. This sensitivity however is affected by the frequency of such events and can change over time. Authors should better discuss the limitations and implications of their work with regards to users' sensitivity. For instance, are the noticeable thresholds changing over time? How? What does this communicate, and what could be its consequences on local planners and decision makers?
- Still related to my previous point, did authors carry out temporal analyses? Since they have access to a largely temporally distributed flood ground truth and to temporally rich social media data, they should also touch this point in their work. We know that many social systems can suffer from issues related to temporal and concept drifts. As such, it would be nice to have some experiments about this point, or at the very least, to discuss possible limitations of the presented approach that are related to temporal and concept drifts.

We thank the reviewers for their thoughtful and constructive comments. We have made a number of changes to the manuscript and believe we have addressed all major concerns, resulting in an improved paper. Reviewer comments are in black below and our responses to each point given in blue. Thank you for considering this revised manuscript.

Reviewer #1 (Remarks to the Author):

General Comments

The contribution is interesting and well-written, although lacking in detail sufficient for publication. Additionally, there are issues with the methodology that would require clarification and perhaps recomputation of results (described below). Fundamentally, as the results are largely congruent with those of existing methods based on minor flood frequency, the work does not reach the level of *extreme importance to scientists in the specific field*. Therefore, my recommendation is that this work is not suitable for publication in Nature Comm., but may be well-received by a specialist journal after addressing some core issues.

Specific Comments

1) I do not recall seeing any normalisation or consideration of the varying number of Twitter users over the period of record.

The number of Twitter users does vary over the sample period. In general there is a steady increase in the population of users in the sample, interrupted by a drop in September 2014, likely associated with a change in Twitter's policy on geolocating tweets. A figure showing this information has been added to the Supplementary Information in the revised manuscript (Supplementary Figure 3).

In both the original and revised analysis, the number of Twitter users by county and time period (ie. weekly or daily) is controlled for in all regressions. This is now more explicitly discussed in the Methods section:

"The sample contains Tweets from over 5 million unique users. The number of Twitter users is steadily increasing over time, with a sharp drop in late 2014 likely associated with a change in Twitter's policy on geolocating Tweets (Supplementary Figure 3). The number of Twitter users is included as a control variable in all regressions."

2) Mention is made of "grasping consequences", "coastal flood preparation and response", yet nowhere is FEMA and it's flood maps mentioned. As those are the ultimate insurance-based reference for "response" to the community, it seems that these maps should be considered.

We thank the reviewer for pointing out this omission. We see the analysis of nuisance coastal flood thresholds in our analysis as largely complementary to standard planning tools such as the FEMA flood maps. FEMA produces maps for both the 1% annual flood frequency and the 0.2% annual flood frequency. Both are both much rarer and much more consequential than the vast majority of the minor flood events in our dataset - most locations experience at least one minor flood event per year (Figure 1a). Therefore, although smaller and less devastating than the flood

events mapped by FEMA, because these nuisance floods occur regularly, and will occur more regularly with sea level rise, understanding their frequency and effects is likely still important for local planners.

We have clarified this point in the text with the following addition:

“As such they can complement existing flood planning tools. In particular, social media might provide a sensitive instrument to measure nuisance coastal flooding that is both more regular and less consequential than the flooding covered by other tools such as the FEMA flood maps of 1 in 100 and 1 in 500 year floodplains.” (Introduction)

3) Line 15:

Flood thresholds based on remarkability can provide useful information to complement existing measures of flood frequency.

-- Thus assertion seems reasonable, but is speculative. A clear quantitative demonstration is needed to support such a statement.

This statement has been replaced with a quantitative statement directly linked to the findings in the paper:

“While thresholds in most counties are statistically-indistinguishable from the existing minor flood thresholds of nearby tide gauges, we find evidence that several areas experience noticeable flooding at a height lower than existing thresholds. These 22 counties include several major cities such as Miami, New York and Boston, and contain a total population of over 13 million. Our analysis implies that large numbers of people might currently be exposed to nuisance flooding not identified from standard tide gauge measures.”

4) Line 65-67, 236-243

Tweets about flooding are classified using a simple “bag-of-words” approach, where any post containing at least one word or phrase identified as possibly referring to flood events was labeled as a flood tweet (see Supplementary Information).

-- I did not find a Supplement, but this seems to be in Methods.

Our apologies to the reviewer - the reference to SI was incorrect. The correct reference to the Methods section has been incorporated into the revised manuscript.

-- The list of words to identify flood-relevant tweets is ad-hoc, and I do not recall any sensitivity study to identify the relative importance and impact of specific, or subsets of words on the results.

While having limitations, using crowd-sourced word lists (bag-of-words approaches) is a common practice of concept identification within the literature. For example, both Coviello et al. (1) and Baylis et al. (2) employ the method to identify weather-related terms in their analyses, and Moore et al. (3), likewise, employ the method. Further, the use of specific terms to identify underlying constructs is incredibly common in the psychological literature. For example, the widely validated and popular tool, LIWC (Linguistic Inquiry and Word Count), employs lists of

words to identify and classify emotional constructs such as anger, cognitive complexity, and happiness (4, 5). The identification of flood-related tweets is substantially less nuanced than these more complex constructs.

That said, lists of words might indeed produce systematic biases of concern to our analyses. To address this concern, we randomly identified three outlier counties and performed a manual validation of all tweets in these counties identified as flood-related using the bag of words omitting the words identified as problematic by the reviewer in the following comment. This produced a dataset of 3305 validated tweets. This analysis demonstrates that the false positive rate is high (63%), but we are also able to show that this error is random, meaning it will introduce noise but not bias into our estimates.

Specifically, aggregating our dataset of true_positive and false_positive tweets to the county * day level, we separately regress the number of true_positive tweets about flooding and the number of false_positive tweets containing flood words on an indicator for whether the maximum tide height was above the gauge flooding threshold, as well as the set of standard controls included in all regressions in the paper (namely quartic in daily precipitation, quadratic in five-day precipitation, the number of Twitter users, and month-of-year and county fixed effects). The analysis shows that the number of true_positive tweets about flooding are closely associated with extreme tide heights, but the number of false_positives are not (see table below). This implies that, conditioned on the set of controls, variation arising from mis-identified tweets will not systematically bias our estimated relationships between tide height and the volume of tweets.

	Flood-Stage Coefficient	Standard Error	p-value
True-Positive Flood Tweets	1.254	0.193	<0.0001
False-Positive Flood Tweets	0.024	0.937	0.980

Table One: Coefficient on an indicator variable for whether daily tide height was above gauge flood threshold with two dependent variables in a negative binomial regression. Top row is the number of tweets about flooding identified as true positives, bottom row is the number of false positive tweets about flooding. Both regressions include controls for daily precipitation (quartic), five day precipitation (quadratic), number of Twitter users, month of year, and county. There is no evidence that the number of false-positive tweets are associated with extreme tide heights, meaning this error will introduce noise but not bias into our estimates. Data is from the complete set of 3305 manually-validated, flood-related tweets from three counties randomly chosen from the set of outlier counties.

This information has been included in the revised manuscript. Note that this analysis focuses on false-positive errors, which are a far larger concern than false-negatives simply because tweets about flooding are a very very small fraction of overall tweets (less than 0.05% in our dataset). In addition, the bag-of-words is designed to be fairly broad, so as to generally generate more false positive than false negatives. Given these priors, the false-negative rate should be

extremely low. This was found in a previous validation of classification of tweets about weather using the same bag-of-words approach reported in Moore et al. (2019) - the false-positive rate was ~ 45% while the false-negative rate was <0.5%.

-- These specific words: [army corps of engineers, levee, levees, levy, deep water, shallow water, saturate, saturated, saturates, soak, soaked, soaks, puddle, coastal, splash, splashed, splashes, spray, sprays, sprayed, got wet, get wet, rising sea, sea-level, sea-level-rise, sea level, SLR]
are too general to be considered flood-specific.

We agree with the reviewer on this point. In the revised manuscript these words have been removed from the bag of words used to identify Tweets about flooding. Our findings are robust to this change. One exception is that a slightly larger set of outlier counties are identified, likely because response functions are more-precisely estimated.

5) Figure 1 is unconvincing as a demonstration of "a steady increase in the average number of flood events over time".

The statement about increasing average number of flood events over time actually referred to Supplementary Figure 1 in the original manuscript. Since this was confusing, the relevant figure has been moved to the main text as Figure 1b and the reference adjusted accordingly.

Reviewer #2 (Remarks to the Author):

This work discusses an approach to the estimation of coastal flooding threshold that is based on an analysis of social media data. In particular, authors learn new local flood threshold based on the increase in number of flood-related tweets. They then compare their thresholds with those typically used and derived from flood gauges. Results show that some differences exist between social media derived and traditional flood thresholds. For some counties, authors claim that the thresholds derived from social communications might be better representatives of the actual impact of flood events. An analysis of the aforementioned differences also highlights some areas where the density of flood gauges should be increased, thus possibly contributing to drive future interventions.

The topic of this work is interesting and timely and the paper is well-written and easy to follow. In addition, the figures are clear, nice, and very informative. From the technical viewpoint however, the work is a bit shallow. Authors could have carried out additional experiments, as detailed in the following, to better evaluate their system/thresholds and avoid possible biases. Please, find below a detailed list of concerns and suggestions:

- The initial claims of the Introduction section (e.g., flooding is increasingly common) would be more convincing with supporting references, in addition to Supplementary Figure 1.

We thank the reviewer for this observation and have added several new references to the introduction.

- An important pioneering reference on the use of social media data for earthquake management, that authors might consider citing, is the following:
Earle, Paul. "Earthquake twitter." *Nature Geoscience* 3.4 (2010): 221.

This important reference has been added to the text.

- Among the works leveraging social media data for assessing the extent of damage, it might be worth citing:
Avvenuti, Marco, et al. "Nowcasting of Earthquake Consequences Using Big Social Data." *IEEE Internet Computing* 6 (2017): 37-45.

We thank the reviewer for drawing our attention to this relevant paper. This citation has been added.

- In explaining results for the noticeable thresholds, it would be very useful to have a geo map of the discrepancy between noticeable thresholds and gauge thresholds (with also markers for gauge positions). This would help to better assess the conclusions drawn by the authors.

We thank the reviewer for this suggestion. The proposed map has been added as a panel to Figure 3 and is discussed in the main text.

- I am also interested in assessing whether the discrepancy might be due to some bias. For instance, is it possible that counties with lower (Twitter) population could result in bigger discrepancies? Authors should account for this factor (and possibly other factors) when presenting their results.

The number of Twitter users varies substantially both between counties and over time. Additional information on the dynamics of the Twitter population in the sample have been added to the main text and supplemental information in response to a comment from Reviewer 1 (Methods and Supplementary Figure 3). Counties with very low populations of Twitter users are largely dropped from the analysis because they do not meet the threshold number of flood-related tweets (at least 20% of days must have at least one tweet identified as flood-related).

More generally however, we thank the reviewer for pointing out this question. Although it is not clear that Twitter population would bias the findings in a particular direction (i.e. lower or higher estimated thresholds), counties with smaller populations and therefore fewer tweets might well have response functions that are less precisely estimated. This might lead to larger variance in the estimated response function and therefore larger discrepancies (in absolute terms) between the estimated and gauge flood thresholds.

We test for this bias in several regressions now reported in Supplementary Table 2. We find no evidence for a relationship between the number of Twitter users in a county and the difference between noticeable and gauge thresholds, either in absolute terms or in terms of quantiles of the tide height distribution. This finding holds both with and without controlling for distance to the nearest tide gauge. (Since rural areas are more likely to be far from tide gauges and to have relatively few Twitter users, this variable is an important control).

We do find evidence that number of Twitter users is associated with the variance of the estimated noticeable flooding threshold (Supplementary Table 2c), consistent with response functions from less populated counties being less precisely estimated due to less data. This

means that rural areas with fewer Twitter users are less likely to be identified as outliers because of the less precisely estimated response functions. These findings and this issue are summarized in Supplementary Table 2 and are discussed in the main text (Discussion and Conclusions)

- From the technical viewpoint, this work is rather simple. Other recent works carried out more detailed analyses on social media-derived data, for example by also considering the content of shared messages to identify expressions of fear or mentions of damage. Interestingly, these works demonstrated the usefulness of such in-depth analyses. Authors of this work however, only take into account the volume of flood-related tweets.

We recognize that previous work using social media data to understand natural disasters has brought other tools to bear, including sentiment analysis and content analysis of posts. Much of this work has been in-depth analysis of single, major events (for example, see Kryvasheyev et al. (6)). There is a trade-off involved between the level of detail and the geographic and temporal scope of analysis. Because of the large geographic and temporal scale of this analysis, which covers over 7,000 possible flood events (measured tide height at the nearest tide gauge was over gauge flood threshold) over 3 years in 236 counties, our methods strike a balance between capturing local heterogeneity through county-level regression models while standardizing the analysis using relatively simple measures of social-media response.

In addition, certain tools such as sentiment analysis are more appropriate for previous work that has focused on major disasters. Many of the events here are more minor and routine events, which may not show a strong sentiment signal in the same way as more major disasters. Textual analysis of flood-related Tweets could be used to analyze the types of impacts reported by residents. This is already done manually in the paper for a single location (Beaumont, TX). Automated textual analysis of Tweets to infer impacts over a larger geographic area is planned for future work but is beyond the scope of this paper.

- In my view, the system proposed in this work is highly dependant on the "sensitivity" of the population in a given area to flood events. This sensitivity however is affected by the frequency of such events and can change over time. Authors should better discuss the limitations and implications of their work with regards to users' sensitivity. For instance, are the noticeable thresholds changing over time? How? What does this communicate, and what could be its consequences on local planners and decision makers?

The authors agree with the reviewer that the noticeability of flooding events may change with repeated exposure. In fact, a previous paper by the authors documents this phenomenon of changing sensitivity for temperature anomalies (see Moore et al. 2019). Unfortunately, the nature of the flooding variation available, combined with the relatively short length of the Twitter dataset makes it more difficult to identify this normalization phenomenon for coastal flooding than for temperature change.

For instance, a critical question of relevance for decision-makers would be whether declining noticeability over time is driven by people becoming desensitized to repeated flooding while still experiencing adverse consequences, or by people adapting so as to be less affected by flooding, thereby making it less noticeable. In the case of temperature, these cases could be distinguished using changes in sentiment, which previous work had shown to be related to

temperature anomalies. Similar work connecting sentiment and minor to moderate flooding events across large geographic areas has not yet been done.

- Still related to my previous point, did authors carry out temporal analyses? Since they have access to a largely temporally distributed flood ground truth and to temporally rich social media data, they should also touch this point in their work. We know that many social systems can suffer from issues related to temporal and concept drifts. As such, it would be nice to have some experiments about this point, or at the very least, to discuss possible limitations of the presented approach that are related to temporal and concept drifts.

We agree with the reviewer that the question of how frequent flooding events do or do not become normalized over time and people become desensitized to the impacts is a fascinating one. As discussed above, however, we do not believe the dataset available allows desensitization to be readily distinguished from adaptation. In other words, places that experience floods infrequently may have a larger response on Twitter than places with more frequent floods either because the floods are surprising and therefore remarkable, or because places with more frequent floods have adjusted so as to make the inundation less consequential and therefore less remarkable. Unfortunately, the length of Twitter dataset (3 years) is not sufficient to resolve changes in the inter-annual frequency of flooding within counties given expected monthly and seasonal variation and natural variability.

To address the reviewer's concern, we have added results from a new analysis in the Supplementary Information as well as a discussion of the issues around temporal dynamics resulting from both normalization and adaptation. The new analysis presents a regression model over the full dataset in which the Twitter to flood stage tide heights is allowed to interact with the number of days since the last flood stage. Normalization and adaptation would both imply that the response on Twitter would increase with the number of days since the last flood, which we do find evidence for (figure reproduced below). This issue is discussed in the revised main text (discussion and conclusions).

Supplementary Figure 2: Response function showing % change in the number of flood-related Tweets over the whole sample on flood days, as a function of number of days since the last flood event (upper panel). Regression is a negative binomial regression at the daily level and includes controls for daily precipitation (quartic), cumulative five-day precipitation (quadratic), the number of Twitter users, as well as county, state-month, and year fixed effects and county-specific time trends. Lower panel shows the frequency of intervals between floods.

REVIEWERS' COMMENTS:

Reviewer #1 (Remarks to the Author):

The authors have diligently addressed all of my concerns. I find the manuscript to be logically complete and suitable for publication at the editors discretion.

Reviewer #2 (Remarks to the Author):

The authors have satisfactorily addressed all my previous concerns and the manuscript has improved much with this revision. As such, I think that this work is ready for publication.

Dr. Stefano Cresci (IIT-CNR)